# In-Context Learning under Distribution Shift: Optimal Attention Temperature for Transformers

## Abstract

Pretrained Transformers exhibit strong in-context learning (ICL) capabilities, enabling them to perform new tasks from a few examples without parameter updates. However, their ICL performance often deteriorates under distribution shifts between pretraining and test-time data. Recent empirical work suggests that adjusting the attention temperature—a scaling factor in the softmax—can improve the performance of Transformers under such distribution shifts, yet its theoretical role remains poorly understood. In this work, we provide the first theoretical analysis of attention temperature in the context of ICL with pretrained Transformers. Focusing on a simplified setting with "linearized softmax" attention, we derive closed-form expressions for the generalization error under distribution shifts. Our analysis reveals that distributional changes in input covariance or label noise can significantly impair ICL, and that an optimal attention temperature exists which provably minimizes this error. We validate our theory through simulations on linear regression tasks and experiments with LLaMA2-7B on question-answering benchmarks. Our results establish attention temperature as a critical lever for robust in-context learning, offering both theoretical insight and practical guidance for tuning pretrained Transformers under distribution shift.

## 1 Introduction

Transformers [27] have become the cornerstone of modern AI systems, powering state-of-the-art models such as ChatGPT, Gemini, and DeepSeek. A key capability underlying their success is *in-context learning* (ICL)—the ability to adapt to new tasks directly from prompts, without modifying internal weights [4]. This emergent behavior has sparked significant interest in understanding the mechanisms behind ICL [2, 29], as well as how factors such as task diversity and model scale influence performance [30, 33].

Despite its promise, ICL remains sensitive to distribution shifts between pretraining and downstream tasks. Empirical and theoretical studies have shown that such shifts can degrade performance [35], raising critical questions about the robustness and adaptability of pretrained Transformers.

At the heart of the Transformer architecture lies the self-attention mechanism, formally expressed as

$$\text{Attention}(\boldsymbol{Z}) := \boldsymbol{V}\boldsymbol{Z} \cdot \text{softmax}\left(\frac{(\boldsymbol{K}\boldsymbol{Z})^T(\boldsymbol{Q}\boldsymbol{Z})}{\tau}\right), \tag{1}$$

where $\boldsymbol{Z}$ is the input, and $\boldsymbol{Q}$, $\boldsymbol{K}$, and $\boldsymbol{V}$ are the query, key, and value weight matrices, respectively. The parameter $\tau > 0$, known as the *attention temperature*, modulates the sharpness of the softmax distribution. While the original Transformer set $\tau = \sqrt{d_k}$ [27] where $d_k$ is the dimension of the key matrix, later works in both NLP and vision have found that tuning or learning attention temperature can improve performance [16, 36, 21, 13, 5, 37].

Submitted to 39th Conference on Neural Information Processing Systems (NeurIPS 2025). Do not distribute.

Temperature controls how sharply attention weights focus on certain inputs—a property that could play a critical role under distribution shift. Surprisingly, despite its operational importance, the effect of temperature on the ICL behavior of pretrained Transformers has received little theoretical attention. This gap is particularly relevant in practice, where distribution mismatch between training and deployment is the norm.

**This work —** In this paper, we present a theoretical and empirical investigation of the attention temperature in the context of ICL. Our main focus is on how tuning temperature can improve the generalization performance of pretrained Transformers under distribution shifts. We study this question in the setting of linear regression tasks, which serve as a tractable framework for understanding ICL [9, 35]. Departing from prior work that considers linear attention, we analyze a Transformer with *linearized softmax* attention, which retains the essential temperature-dependent behavior of standard attention while allowing for mathematical tractability.

Our analysis identifies a closed-form expression for the *optimal temperature*—the value of $\tau$ that minimizes generalization error during inference. We show that this optimal temperature depends explicitly on the nature of the distribution shift, and that setting it appropriately can recover or even surpass baseline ICL performance. We validate our theoretical predictions through extensive experiments on both synthetic (linear regression) and real-world (question answering with LLMs) tasks, demonstrating that temperature tuning offers a simple yet powerful mechanism to improve robustness.

**Contributions —** Our work makes the following contributions:

1. We theoretically characterize the optimal attention temperature for pretrained Transformers with linearized softmax attention in in-context learning tasks.
2. We analyze the generalization behavior of such models under a broad range of distribution shifts, using a relaxed set of assumptions compared to prior work.
3. We establish a clear theoretical and empirical link between distribution shifts and temperature, showing that tuning temperature significantly enhances ICL performance across tasks.

Taken together, our results offer new insights into the interplay between temperature, distribution shift, and generalization in in-context learning, with implications for both theory and practice in the deployment of pretrained Transformers.

## 2 Related work

**In-context learning —** The ICL capability of Transformers was first brought to prominence by [4], leading to a surge of empirical and theoretical investigations. Several works have demonstrated that ICL performance improves with model scale [30, 19, 25], underscoring its importance in modern AI systems.

To better understand this phenomenon, synthetic tasks such as linear regression have served as controlled testbeds for analyzing ICL in Transformers [9, 35, 24]. A prevailing hypothesis in recent theoretical work is that Transformers implicitly learn algorithms during pretraining, which they subsequently execute during inference [3, 14, 2, 1, 29, 18, 7, 35, 15, 20]. However, there remains ongoing debate over the precise nature of these learned procedures.

In this context, simplified Transformer variants—particularly those using linear attention—have proven useful for gaining analytical insights. Notably, [35] showed that linear Transformers approximate Bayes-optimal inference in linear regression tasks, even under distribution shift.

Our work builds on this line of research but focuses specifically on the role of the temperature parameter in attention. Unlike [35], we (i) employ linearized softmax attention to isolate the influence of temperature, (ii) study how temperature adjustments can mitigate the effects of distribution shifts, and (iii) derive and evaluate the optimal temperature for improving ICL performance. These contributions extend prior analyses and provide a deeper understanding of how tuning temperature can enhance Transformer generalization under distributional variations.

**Linear vs. softmax attention —** A parallel thread of research investigates the comparative efficacy of linear and softmax attention mechanisms, which is directly relevant to our study since temperature is traditionally associated with softmax attention. While linear attention has gained popularity for its computational efficiency, it is often outperformed by softmax-based counterparts, prompting efforts to close this performance gap [6, 22].

A key development in this area is the work of [11], who demonstrated that a linearized variant of softmax attention can closely match the performance of standard softmax attention. Motivated by this finding, we adopt the "linearized softmax" formulation, allowing for tractable theoretical analysis while preserving the critical role of temperature. This approach facilitates a principled investigation of how temperature tuning impacts ICL in pretrained Transformers.

**Temperature —** Despite its central role in attention mechanisms, the temperature parameter remains underexplored in the context of ICL. Recent work by [28] proposes adaptive temperature as a means to sharpen softmax outputs, and temperature adjustments are sometimes reported in empirical studies of pretrained LLMs [26]. However, a systematic analysis of temperature's effect on ICL—particularly under distribution shift—has been lacking.

To address this gap, we provide a theoretical and empirical investigation of temperature within Transformers using "linearized softmax" attention. Our results clarify how the optimal temperature depends on the data distribution and how it can be tuned to reduce generalization error in in-context learning scenarios.

# 3 Setting

We describe the setup for analyzing ICL in linear regression using pretrained Transformers, covering the data model, linearized attention with reparameterization, evaluation metrics, and the Bayes-optimal benchmark.

**Notation —** We follow standard notation from [10]. The spectral norm of matrix $M$ is denoted by $\|M\|$, and the trace by $\text{Tr}(M)$. Matrix entries and slices are denoted as $M_{i,j}$, $M_{:,j}$, and $M_{i,:}$.

## 3.1 Problem setup: In-context learning for linear regression

We study the ICL abilities of pretrained Transformers on linear regression tasks. Given a sequence of tokens, i.e., input-label pairs, $\{x_1, y_1, x_2, y_2, \ldots, x_{l-1}, y_{l-1}, x_l\}$, where each input vector $x_i \in \mathbb{R}^d$ and corresponding label $y_i \in \mathbb{R}$ are independently sampled from an unknown joint distribution, the model must predict $y_l$ using only the context $\{(x_i, y_i)\}_{i=1}^{l-1}$ and the query $x_l$, where $l - 1$ is referred as the "context length". Each $(x_i, y_i)$ pair is sampled i.i.d. from a joint distribution defined by:

$$x_i \sim \mathcal{N}(\boldsymbol{\mu}_x, \boldsymbol{\Sigma}_x), \quad y_i = \boldsymbol{w}^T x_i + \epsilon_i, \quad \epsilon_i \sim \mathcal{N}(0, \sigma^2), \tag{2}$$

where the task vector $\boldsymbol{w} \sim \mathcal{N}(\boldsymbol{\mu}_w, \boldsymbol{\Sigma}_w)$ is fixed within a context but varies across tasks.

**Assumption 3.1** (Well-Behaved Data Distributions). There exist constants $c_1, c_2, c_3 > 0$ such that:

$$\|\boldsymbol{\mu}_x\|, \|\boldsymbol{\mu}_w\| \leq c_1, \quad \lambda_{\min}(\boldsymbol{\Sigma}_x), \lambda_{\min}(\boldsymbol{\Sigma}_w) \geq c_2, \quad \lambda_{\max}(\boldsymbol{\Sigma}_x), \lambda_{\max}(\boldsymbol{\Sigma}_w) \leq c_3.$$

This assumption ensures well-behaved distributions by bounding the means and covariances of input and task vectors, offering greater flexibility than the more restrictive setup in [35].

**Assumption 3.2** (High-Dimensional Regime). The context length $l$ and input dimension $d$ diverge jointly: $l, d \to \infty$.

This assumption reflects realistic settings where both context length and input dimension grow, aligning with modern ML trends and enabling analysis of generalization in high-dimensional regimes.

Under this set of assumptions, we define ICL for linear regression tasks as follows:

**Definition 3.3** (In-Context Learning (ICL)). A model succeeds at ICL for linear regression if its generalization error nearly matches that of the Bayes-optimal linear model (defined in Section 3.6).

## 3.2 Modeling attention with transformers

Following the convention established in [35], we embed the input sequence into an embedding matrix:

$$Z := \begin{bmatrix} x_1 & \cdots & x_{l-1} & x_l \\ y_1 & \cdots & y_{l-1} & 0 \end{bmatrix} \in \mathbb{R}^{(d+1) \times l}, \tag{3}$$

where the last column corresponds to the query input with no label.

127  Using the embedding matrix, the softmax self-attention output is given by:

$$\boldsymbol{S} := \boldsymbol{Z} + \boldsymbol{V}\boldsymbol{Z} \cdot \text{softmax}\left(\frac{(\boldsymbol{K}\boldsymbol{Z})^T(\boldsymbol{Q}\boldsymbol{Z})}{\tau}\right), \tag{4}$$

128  where $\boldsymbol{K}$, $\boldsymbol{Q}$, and $\boldsymbol{V}$ are the key, query, and value matrices, respectively, and $\tau$ is the temperature.

129  Here, we denote the model's prediction as $S_{d+1,l}$ — the last element in the final row.

### 3.3  Linearized attention approximation

131  To analytically study the effect of temperature on ICL, we adopt a linearized approximation of
132  softmax attention (see Appendix B for the derivation):

$$\boldsymbol{E} := \boldsymbol{Z} + \frac{1}{l}\boldsymbol{V}\boldsymbol{Z}\left(\frac{(\boldsymbol{K}\boldsymbol{Z})^T(\boldsymbol{Q}\boldsymbol{Z})}{\tau} + \boldsymbol{1} - \frac{1}{l}\sum_{j=1}^{l}\frac{(\boldsymbol{K}\boldsymbol{Z}_{:,j})^T(\boldsymbol{Q}\boldsymbol{Z})}{\tau}\right), \tag{5}$$

133  where $\hat{y} := E_{d+1,l}$ is the predicted label. Unlike traditional linear attention (e.g., [35]):

$$\boldsymbol{Z} + \frac{1}{l}\boldsymbol{V}\boldsymbol{Z}(\boldsymbol{K}\boldsymbol{Z})^T(\boldsymbol{Q}\boldsymbol{Z}), \tag{6}$$

134  our linearized version preserves normalization properties, improving interpretability and robustness.

135  *Remark* 3.4 (Linear vs. Linearized Attention). Linearized attention preserves row-wise normalization,
136  making it more robust to variations in input means — a key failure mode of linear attention in ICL.
137  Appendix C illustrates this distinction.

### 3.4  Reparametrization of linearized attention

139  To streamline analysis, we reparametrize the matrices $\boldsymbol{V}$ and $\boldsymbol{M} := \boldsymbol{K}^T\boldsymbol{Q}$ as:

$$\boldsymbol{V} = \begin{bmatrix} * & * \\ \boldsymbol{v}_{21}^T & v_{22} \end{bmatrix}, \quad \boldsymbol{M} = \begin{bmatrix} \boldsymbol{M}_{11} & * \\ \boldsymbol{m}_{21}^T & * \end{bmatrix}, \tag{7}$$

140  where only $\boldsymbol{v}_{21}$, $v_{22}$, $\boldsymbol{m}_{21}$, and $\boldsymbol{M}_{11}$ influence the prediction $\hat{y}(\boldsymbol{Z}; \boldsymbol{V}, \boldsymbol{M})$. The remaining terms
141  are denoted by $*$ as they are not relevant for predicting $y_l$ in this context. The prediction from the
142  linearized attention model can thus be expressed as a function of $\boldsymbol{M}$ and $\boldsymbol{V}$, i.e., $\hat{y}(\boldsymbol{Z}; \boldsymbol{V}, \boldsymbol{M}) :=$
143  $E_{d+1,l}$. This form parallels the approach in [35], allowing for tractable theoretical analysis.

144  By analyzing this reparameterization, we gain a deeper understanding of how the model parameters
145  interact with the data to address the ICL problem effectively. This foundational insight will provide
146  the necessary basis for discussing the pretraining of these parameters in Section 4.1.

### 3.5  Evaluating generalization performance

148  We focus on evaluating the performance of our attention model by assessing its generalization error.
149  For a given set of parameters $(\boldsymbol{V}, \boldsymbol{M})$, the model's generalization (ICL) error is:

$$\mathcal{G}(\boldsymbol{V}, \boldsymbol{M}) := \mathbb{E}_{(\boldsymbol{Z}, y_l) \sim \mathcal{D}^{test}}\left[(y_l - \hat{y}(\boldsymbol{Z}; \boldsymbol{V}, \boldsymbol{M}))^2\right], \tag{8}$$

150  where $\mathcal{D}^{test}$ denotes the distribution of the test set, which includes input-output pairs that the model
151  has not encountered during training. In this context, the ICL task assesses the genuine ICL capabilities
152  of the linearized attention module. Here, the task vectors in the test set differ from those encountered
153  during training, requiring the model to infer these new vectors based solely on the provided context.

### 3.6  Bayes-optimal ridge estimator

155  The Bayes-optimal ridge estimator provides a robust framework for estimating the task vector $\boldsymbol{w}$
156  given a prior distribution and a set of $l-1$ samples. It is defined as:

$$\hat{\boldsymbol{w}}_{Bayes} = \left(\frac{\bar{\boldsymbol{X}}^T\bar{\boldsymbol{X}}}{\sigma^2} + \boldsymbol{\Sigma}_w^{-1}\right)^{-1}\left(\frac{\bar{\boldsymbol{X}}^T\bar{\boldsymbol{y}}}{\sigma^2} + \boldsymbol{\Sigma}_w^{-1}\boldsymbol{\mu}_w\right), \tag{9}$$

where $\bar{X}$ is the centered input matrix and $\bar{y}$ is the centered label vector. This estimator integrates data information while incorporating prior beliefs about the distribution of $w$, effectively balancing bias and variance, hence serves as the gold standard against which we compare model predictions. The terms including $\Sigma_w^{-1}$ introduce a regularization effect, which is especially beneficial in high-dimensional settings.

The derivation of this estimator, detailed in Appendix A, illustrates how Bayesian principles can inform regression techniques by combining observed data with prior distributions to yield more reliable predictions. In our context, the inputs and labels originate from the prompt matrix $Z$, and the prediction of the Bayes-optimal linear model for any input $x$ is given by $\hat{w}_{Bayes}^T x$.

# 4 Theoretical results

In this section, we present our main theoretical results on the behavior of the linearized attention model in the context of ICL. We begin by showing how to pretrain the model to approximate the Bayes-optimal linear predictor, thereby grounding its predictive performance. We then identify specific conditions under which the model fails to generalize under distribution shifts at test time, revealing key limitations of linearized attention in ICL. Following this, we provide a detailed characterization of its generalization error, offering a principled framework for analyzing performance. Finally, we investigate the role of the temperature parameter and demonstrate that tuning it appropriately can substantially improve generalization—especially in cases where the model initially fails to perform effective in-context learning.

## 4.1 Model pretraining

We begin our pretraining analysis by observing that the prediction generated by the linearized attention model can be reduced to the following form (see Appendix D for the derivation):

$$\hat{y}(Z; V, M) := E_{d+1,l} = \frac{1}{\tau} \hat{w}_{Att}(C_{xx}, C_{xy}, C_{yy}; M, V)^T x_l + b_{Att}(s_x, s_y; V), \quad (10)$$

where $\hat{w}_{Att}(C_{xx}, C_{xy}, C_{yy}; M, V) \in \mathbb{R}^d$ and $b_{Att}(s_x, s_y; V) \in \mathbb{R}$. $s_x$ and $s_y$ denote the sample means of the input $x$ and the label $y$, respectively, and $C_{xx}$ and $C_{xy}$ are the sample covariances corresponding to $\text{Cov}(x)$ and $\text{Cov}(x, y)$. These statistics are computed from the prompt matrix $Z$.

For pretraining, we optimize the parameters $V$ and $M$ using $m$ samples of $(Z, y_l)$ drawn from the distribution $\mathcal{D}^{train}$, where each $Z$ contains $l - 1$ $(x, y)$ pairs intended for ICL. Building upon prior work that connects ICL in linear regression to the Bayes-optimal ridge estimator [35, 24], we configure $M$ and $V$ to emulate Bayes-optimal ridge regression. Specifically, we aim for $\hat{w}_{Att}(C_{xx}, C_{xy}; M, V) \approx \hat{w}_{Bayes}$ and $b_{Att}(s_x, s_y; V) \approx 0$.

**Lemma 4.1** (Pretrained Parameters). *When the temperature parameter is set to $\tau = 1$ during pretraining, the following parameter configuration approximates the Bayes-optimal estimator in (9):*

$$M_{11} = d\left(\frac{\hat{X}^T \hat{X}}{ml} + \frac{\sigma^2}{l} \Sigma_w^{-1}\right)^{-1}, \quad m_{21} = 0, \quad (11)$$

$$v_{21} = \frac{\sigma^2}{dl}\left(\frac{\hat{X}^T \hat{X}}{ml}\right)^{-1} \Sigma_w^{-1} \mu_w, \quad v_{22} = \frac{1}{d},$$

*where $\hat{X} \in \mathbb{R}^{ml \times d}$ is the centered input matrix formed from $ml$ samples of $x$. This configuration aligns the linearized attention model with Bayes-optimal ridge regression. The quantities $\mu_w$ and $\Sigma_w$ can be estimated from the pretraining data. A detailed derivation is provided in Appendix E.*

This lemma establishes a theoretical connection between the pretrained parameters and the Bayes-optimal estimator, reinforcing the foundation of our approach.

Moreover, specific instances of Lemma 4.1 recover settings explored in prior studies. For example, under the assumptions $\Sigma_x = \Sigma_w = I$, $\mu_w = 0$, and $\sigma = 0$, [29] employ $M_{11} = \text{Cov}(x)^{-1}$ and $v_{21} = 0$ within a linear attention framework. Our formulation generalizes this by allowing $v_{21} \neq 0$, which reflects our assumption that $\mu_w \neq 0$—a departure from earlier works. In our self-attention-based analysis, $v_{21}$ encodes task vector bias. Additionally, our choice of $M_{11}$ explicitly

accounts for label noise ($\sigma^2$), thereby enhancing the model's adaptability and maintaining a Bayesian interpretation.

We further comment on task diversity and parameter optimality in the following two remarks:

*Remark* 4.2. A high degree of task diversity (i.e., the number of distinct tasks) is crucial for enabling in-context learning [33]. In our framework, task diversity significantly affects the accuracy of estimating $\boldsymbol{\mu}_w$ and $\boldsymbol{\Sigma}_w$ during pretraining.

*Remark* 4.3. Although the pretrained parameters specified in Lemma 4.1 may not be optimal in all scenarios, they are analytically valuable for understanding the effects of distribution shifts and the influence of the temperature parameter in ICL. Notably, our characterization of ICL performance and temperature optimality does not rely on these specific parameter choices.

Based on Lemma 4.1, we arrive at the following corollary:

**Corollary 4.4.** *Suppose there is no distribution shift between training and inference. Then, under the parameter configuration of Lemma 4.1, the linearized attention model emulates the Bayes-optimal linear model, implying that it is capable of in-context learning according to Definition 3.3.*

Since the pretrained model succeeds in ICL for $\mathcal{D}^{test} = \mathcal{D}^{train}$, we next investigate how distribution shifts affect its ICL performance.

## 4.2 Effect of distribution shift

In this section, we explore scenarios where $\mathcal{D}^{test} \neq \mathcal{D}^{train}$, indicating a shift in the input, task, or noise distribution after pretraining the linearized attention model. We consider three cases: (1) a shift in the input distribution (altered mean or covariance), (2) a shift in the task distribution, and (3) a change in the noise levels.

To evaluate the impact of these distribution shifts on ICL performance, we assess whether adjustments to $\boldsymbol{M}$ and/or $\boldsymbol{V}$ are necessary to match the Bayes-optimal linear model under the new distribution. If so, the model is considered sensitive to the shift. Otherwise, it is deemed robust.

**Case I: Shift in input distribution —** Recall that inputs are drawn as $\boldsymbol{x}_i \sim \mathcal{N}(\boldsymbol{\mu}_x, \boldsymbol{\Sigma}_x)$, as defined in (2). Let $\boldsymbol{\mu}_x^{train}, \boldsymbol{\Sigma}_x^{train}$ and $\boldsymbol{\mu}_x^{test}, \boldsymbol{\Sigma}_x^{test}$ denote the input means and covariances for pretraining and testing, respectively. We consider two subcases:

(i) **Shift in mean ($\boldsymbol{\mu}_x^{train} \neq \boldsymbol{\mu}_x^{test}$):** The mean shift does not affect the linearized attention model since it uses centered inputs. However, this impacts the linear attention model, which operates on uncentered inputs, as discussed in Remark 3.4.

(ii) **Shift in covariance ($\boldsymbol{\Sigma}_x^{train} \neq \boldsymbol{\Sigma}_x^{test}$):** A covariance shift necessitates retraining, as $\boldsymbol{M}_{11}$ is tailored to the pretraining input covariance. A mismatch leads to a significant deviation from the Bayes-optimal estimator, consistent with findings in prior work on linear attention [35].

**Case II: Shift in Task Distribution —** The task vectors follow $\boldsymbol{w} \sim \mathcal{N}(\boldsymbol{\mu}_w, \boldsymbol{\Sigma}_w)$. Let $\boldsymbol{\mu}_w^{train}, \boldsymbol{\Sigma}_w^{train}$ and $\boldsymbol{\mu}_w^{test}, \boldsymbol{\Sigma}_w^{test}$ be the mean and covariance of the task distribution during pretraining and testing, respectively. The linearized attention model can incorporate $\boldsymbol{\mu}_w^{train}$ and $\boldsymbol{\Sigma}_w^{train}$ via the pretrained parameters $\boldsymbol{M}_{11}$ and $\boldsymbol{v}_{21}$ (see Lemma 4.1). However, as the context length $l$ increases, the model's dependence on the task distribution diminishes. Thus, shifts in the task distribution primarily affect ICL performance for small $l$.

**Case III: Shift in noise distribution —** Finally, consider a change in the noise distribution: $\epsilon_i \sim \mathcal{N}(0, \sigma^2)$, with $\sigma_{train}^2$ and $\sigma_{test}^2$ denoting pretraining and testing noise variances. If $\sigma_{train}^2 \neq \sigma_{test}^2$, the parameters $\boldsymbol{M}_{11}$ and $\boldsymbol{v}_{21}$ become suboptimal relative to the Bayes-optimal linear model. However, as with the task distribution, the impact of noise shift diminishes as $l \to \infty$.

**Summary —** The linearized attention model is robust to shifts in input mean but sensitive to input covariance changes. Shifts in task or noise distribution reduce ICL performance at small $l$, though increasing $l$ mitigates these effects. In Section 4.4, we explore optimal temperature selection as a way to enhance robustness. First, we analyze the generalization error of the linearized attention model in the next section.

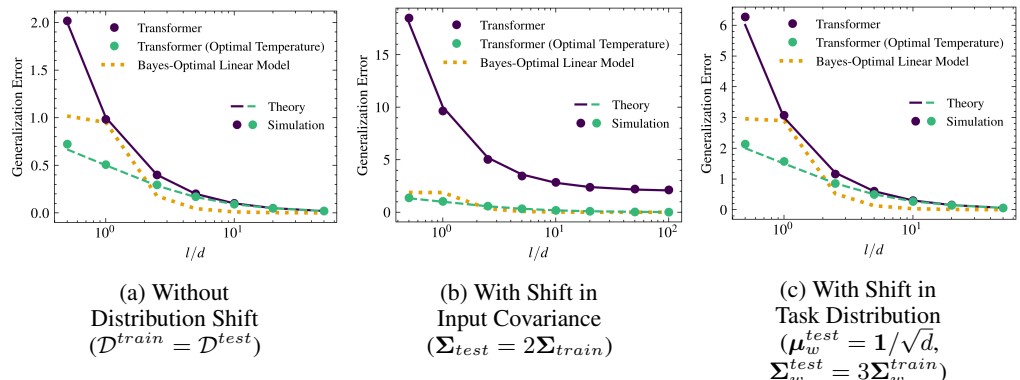

Figure 1: Experiments with Transformer (Linearized Attention) on ICL under distribution shifts. Parameters are set using (11). Here, $d = 50$, $m = 5000$ (a new task per sample), $\sigma = 0.1$, $\boldsymbol{\mu}_x^{train} = \boldsymbol{\mu}_w^{train} = \mathbf{0}$, and $\boldsymbol{\Sigma}_x^{train} = \boldsymbol{\Sigma}_w^{train} = \boldsymbol{I}$.

### 4.3 In-context learning performance

We analyze the in-context learning (ICL) performance of the linearized attention model by evaluating the generalization error defined in (8). To establish a general setting for the subsequent results, we impose the following assumption on the pretrained parameters:

**Assumption 4.5.** There exists a constant $c > 0$ such that

$$\|\boldsymbol{M}_{11}\| \leq cd, \quad \|\boldsymbol{m}_{21}\| = 0, \quad \|\boldsymbol{v}_{21}\| \leq \frac{c}{dl}, \quad |v_{22}| \leq \frac{c}{d}.$$

Note that the pretrained parameters obtained in Lemma 4.1 satisfy Assumption 4.5 with high probability under Assumptions 3.1–3.2. However, the generalization error result stated below holds for any parameters $\boldsymbol{M}, \boldsymbol{V}$ that satisfy Assumption 4.5.

**Theorem 4.6** (Generalization error for ICL). *Suppose Assumptions 3.1–3.2 and 4.5 hold. At test time, assume the input, task, and noise distributions are given by $\mathcal{N}(\boldsymbol{\mu}_x, \boldsymbol{\Sigma}_x)$, $\mathcal{N}(\boldsymbol{\mu}_w, \boldsymbol{\Sigma}_w)$, and $\mathcal{N}(0, \sigma^2)$, respectively. Define*

$$\boldsymbol{A} := \boldsymbol{\Sigma}_x + \boldsymbol{\mu}_x \boldsymbol{\mu}_x^T, \quad \boldsymbol{B} := \boldsymbol{\Sigma}_w + \boldsymbol{\mu}_w \boldsymbol{\mu}_w^T.$$

*Then, the generalization error is*

$$\mathcal{G}(\boldsymbol{V}, \boldsymbol{M}) = \frac{1}{\tau^2} \operatorname{Tr}\left(\boldsymbol{A}\boldsymbol{M}_{11}^T \boldsymbol{F}_1 \boldsymbol{M}_{11}\right) - \frac{1}{\tau} \operatorname{Tr}\left(\boldsymbol{A}\left(\boldsymbol{F}_2 \boldsymbol{M}_{11} + \boldsymbol{M}_{11}^T \boldsymbol{F}_2^T\right)\right) + \operatorname{Tr}\left(\boldsymbol{A}\boldsymbol{B}\right) + \sigma^2, \quad (12)$$

*where*

$$\boldsymbol{F}_1 := \left(\boldsymbol{\Sigma}_x \hat{\boldsymbol{B}} + \frac{1}{l}\left(v_{22}^2 \sigma^2 + \operatorname{Tr}(\hat{\boldsymbol{B}}\boldsymbol{\Sigma}_x)\right)\boldsymbol{I}\right)\boldsymbol{\Sigma}_x, \quad (13)$$

$$\boldsymbol{F}_2 := (\boldsymbol{\mu}_w \boldsymbol{v}_{21}^T + v_{22}\boldsymbol{B})\boldsymbol{\Sigma}_x, \quad (14)$$

*and*

$$\hat{\boldsymbol{B}} := v_{22}\boldsymbol{\mu}_w \boldsymbol{v}_{21}^T + v_{22}\boldsymbol{v}_{21}\boldsymbol{\mu}_w^T + v_{22}^2 \boldsymbol{B}.$$

*Proof.* The generalization error is derived using Isserlis' theorem [12] to compute higher-order moments. See Appendix F for the full derivation. $\square$

Theorem 4.6 illustrates how the parameters $\boldsymbol{M}$, $\boldsymbol{V}$, and the test-time data distribution affect the generalization error. Notably, the temperature parameter $\tau$ plays a critical role.

Although temperature can be implicitly encoded in $\boldsymbol{M}$ during pretraining, it becomes especially important under distribution shifts that the model is not equipped to handle. In such cases, one can optimize generalization performance by choosing the temperature $\tau_{\text{optimal}}$ that minimizes the generalization error, as discussed next.

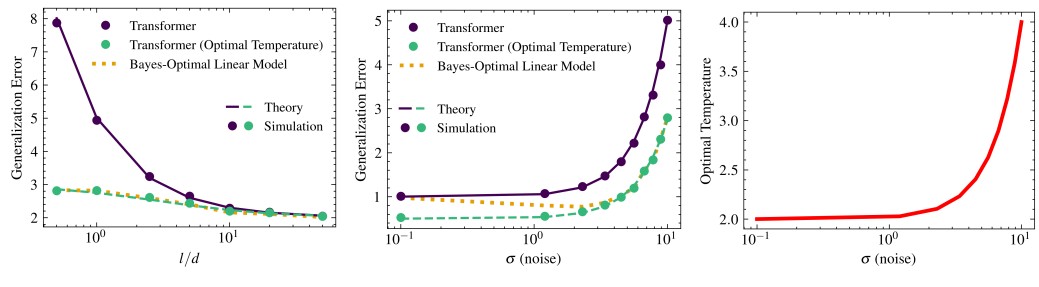

(a) Effect of $l/d$ when $\sigma_{test} = 10$    (b) Effect of $\sigma_{test}$ when $l/d = 1$    (c) Optimal temperature

Figure 2: Effect of noise shift on Transformer (Linearized Attention). The pretraining noise is $\sigma_{train} = 0.1$, while $\sigma_{test}$ varies across plots. Panels (b) and (c) show generalization error and optimal temperature, respectively, as informed by Theorem 4.7. This setting matches Figure 1a, except for changes in test-time noise $\sigma_{test}$.

### 4.4 Optimal attention temperature improves performance

To address distribution shifts, we define the optimal attention temperature as follows:

**Theorem 4.7** (Optimal attention temperature). *Suppose Assumptions 3.1, 3.2, and 4.5 hold. To minimize the generalization error, the optimal attention temperature for inference is given by*

$$\tau_{optimal} = \frac{2Tr\left(\boldsymbol{A}\boldsymbol{M}_{11}^T\boldsymbol{F}_1\boldsymbol{M}_{11}\right)}{Tr\left(\boldsymbol{A}\left(\boldsymbol{F}_2\boldsymbol{M}_{11} + \boldsymbol{M}_{11}^T\boldsymbol{F}_2^T\right)\right)}, \tag{15}$$

*provided that* $Tr\left(\boldsymbol{A}\left(\boldsymbol{F}_2\boldsymbol{M}_{11} + \boldsymbol{M}_{11}^T\boldsymbol{F}_2^T\right)\right) > 0$ *and* $Tr\left(\boldsymbol{A}\boldsymbol{M}_{11}^T\boldsymbol{F}_1\boldsymbol{M}_{11}\right) > 0$.

*Proof.* We minimize the generalization error from Theorem 4.6 with respect to $\tau$ (Appendix H). □

Consider the optimal temperature $\tau_{\text{optimal}}$ from Theorem 4.7. When $\tau_{\text{optimal}} \neq 1$, using an unadjusted temperature leads to suboptimal generalization error. Thus, incorporating the optimal temperature improves generalization in in-context learning under distribution shift.

A natural question is whether the optimal temperature can completely mitigate the adverse effects of distribution shifts. This depends on both the pretraining and test distributions. In some settings, the adjustment can entirely compensate for the shift. For example, if the task distribution is fixed as $\boldsymbol{w} \sim \mathcal{N}(\boldsymbol{0}, \boldsymbol{I})$, the noise variance is $\sigma = 0$, and the input distribution changes from $\boldsymbol{x}_{\text{train}} \sim \mathcal{N}(\boldsymbol{0}, \boldsymbol{I})$ to $\boldsymbol{x}_{\text{test}} \sim \mathcal{N}(\boldsymbol{0}, c\boldsymbol{I})$, then the optimal temperature $\tau_{\text{optimal}} = c$ fully counteracts the shift. In more complex scenarios, it may only partially mitigate the impact, yet still yields improved ICL.

## 5 Experimental results

In this section, we empirically validate our theoretical predictions and demonstrate the impact of the optimal attention temperature on generalization. We begin with controlled experiments on linear regression tasks and progress to evaluating large-scale pretrained models on real-world datasets.

We experiment with two model classes on the linear regression tasks: (i) the linearized attention model, and (ii) the GPT-2 model [23], which incorporates multi-head softmax attention and MLP layers [1]. These experiments show that our theoretical insights generalize from simplified models to more expressive architectures. Finally, we examine the role of temperature in large language models (LLMs), using the Llama2-7B [26] on in-context learning tasks derived from SCIQ dataset [31].

### 5.1 Experiments on linear regression tasks

We consider a Transformer architecture with linearized attention and no MLP layers, as analyzed in our theoretical development. Figures 1 and 2 illustrate its behavior on linear regression tasks (2). Theoretical predictions closely match empirical performance across a range of conditions, confirming the robustness of our analysis. In Figure 1, we compare the ICL performance of the model with and without applying the optimal temperature. As context length $l$ increases (Figure 1a), the

---

[1]Due to the space limitation, we provide the results with GPT-2 in the appendix.

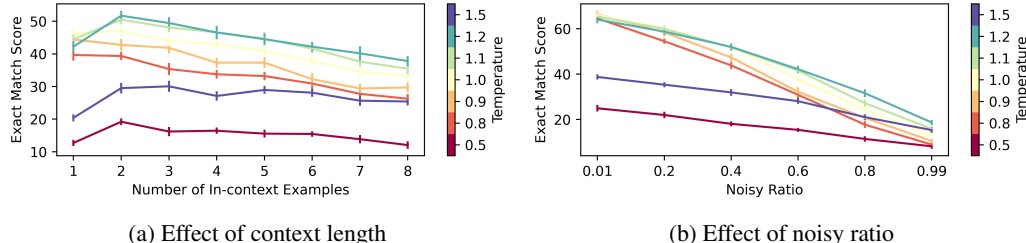

(a) Effect of context length
(b) Effect of noisy ratio

Figure 3: LLM Experiments: The effect of the attention temperature on the ICL performance of the Llama2-7B model [26] using the SCIQ dataset [31]. A distribution shift is introduced by corrupting in-context demonstrations with noisy labels, selected as "relevant" but not necessarily correct answers following [8]. In (a), the noisy ratio is fixed at 0.6; in (b), the number of in-context examples is fixed at 6. Results are averaged over 20 Monte Carlo runs, with error bars indicating 0.33 standard deviations. Attention temperature in all layers is set to $\tau\sqrt{d_k}$, where $d_k$ is the key dimension, to make $\tau$ values dimension-independent. Full details are provided in Appendix I.

model's predictions converge to those of the Bayes-optimal linear model, validating its ICL capability. Figure 1b shows that under an input covariance shift, model performance degrades—but applying the optimal temperature restores alignment with the Bayes-optimal solution. Additionally, Figure 1c shows that the influence of task distribution shift decreases as $l$ increases.

We further evaluate robustness to label noise in Figure 2. In Figure 2a, we observe that noise effects diminish as the context length increases, consistent with our theoretical predictions. However, at small $l$, temperature adjustment becomes critical. In Figure 2b (for $l = d$), the Transformer increasingly diverges from the Bayes-optimal model as noise grows, yet optimal temperature correction closes this gap. Figure 2c shows that the optimal temperature increases with noise level, indicating a principled relationship between noise and temperature under limited context.

## 5.2 Experiments with LLMs for in-context question answering tasks

To assess the practical relevance of our theoretical framework, we investigate how attention temperature impacts the ICL behavior of LLMs. Following prior work [8], we use the SCIQ dataset [31] to create ICL tasks that incorporate distribution shift via noisy labels in the demonstrations. Examples of ICL prompts and the design of noisy labels are provided with full experimental details in Appendix I. We employ the Llama2-7B model [26], evaluating its ICL performance using the exact match score.

Figure 3 presents our results. In Figure 3a, we plot performance as a function of the number of in-context examples under a fixed noisy ratio. Due to the label noise, the performance curve exhibits non-monotonic behavior—highlighting the trade-off between additional context and accumulated noise. Figure 3b shows that as the proportion of noisy demonstrations increases, the optimal temperature also increases, aligning precisely with our theoretical expectations (cf. Figure 2c).

These results affirm that even for highly overparameterized practical models such as Llama2-7B, tuning the attention temperature serves as a principled and effective mechanism to mitigate the negative effects of distribution shifts on in-context learning.

## 6 Conclusion

This work provides a theoretical and empirical foundation for understanding the role of attention temperature in the in-context learning (ICL) capabilities of pretrained Transformers under distribution shifts. By introducing a simplified yet expressive framework based on linearized softmax attention, we analytically characterized how shifts in input covariance and label noise degrade ICL performance. Crucially, we identified and derived an optimal temperature that provably minimizes generalization error in these settings. Our theoretical predictions are validated through extensive experiments on both synthetic linear regression tasks and real-world benchmarks using GPT-2 and LLaMA2 models. Together, our findings offer actionable insights: tuning attention temperature is not merely a heuristic but a principled lever to enhance the robustness of ICL in pretrained Transformers. This advances our understanding of Transformer behavior under distribution shift and opens new directions for improving adaptability in large-scale models.

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
