# OpenReview forum: "In-Context Learning under Distribution Shift: Optimal Attention Temperature for Transformers"
_NeurIPS.cc/2025/Conference — Submitted to NeurIPS 2025_

### Official Review · Reviewer_PaBX · 2025-06-27

**Clarity:** 2
**Significance:** 3
**Originality:** 3
**Rating:** 5
**Confidence:** 3

**Summary:**

The paper analyzes the role of the softmax temperature in the attention mechanism for in-context learning (ICL). It confirms previous empirical findings that scaling the softmax can improve ICL under distribution shifts theoretically. More specifically, the authors find that changes in input covariance or increased label noise can impair ICL performance, and they prove that an optimal temperature exists to minimize this error. For this optimal temperature, the authors derive a closed-form solution using a linearized softmax attention. They evaluate their results on synthetic linear regression tasks and show that the derived optimal temperature performs comparably to the Bayes-optimal linear model. On real data, they demonstrate that ICL performance under noise can be slightly improved by adjusting the temperature. However, in this case, the authors do not use their derived solution to estimate the optimal temperature.

**Questions:**

### **Key Questions and Suggestions for the Authors**

1. **Clarify the Effect of Distribution Shifts on the Softmax Input Distribution:**
   - How exactly do changes in label noise or input covariance influence the **mean and variance of the dot product** `KZᵀ x QZ`?
   - As outlined by Vaswani et al. (2017), the softmax temperature is motivated by the assumption that `KZ` and `QZ` are zero-mean, unit-variance random variables. Do the distribution shifts you study affect these assumptions?
   - Could you empirically estimate the variance of `KZᵀ x QZ` under the shifted distributions you analyze?
   - **Suggestion:** Add the temperature value computed from the **empirical variance of `KZᵀ x QZ`** to Figures 1 and 2, so readers can compare this baseline with your theoretically derived optimal temperature.

2. **Relate Derived Optimal Temperature to Practical Baselines:**
   - Does your closed-form solution for the optimal temperature **deviate significantly** from the empirical variance of `KZᵀ x QZ`?
   - This would help determine whether your theoretical contribution offers **substantial improvement over standard heuristics** already used in practice, for instance see [1].

3. **Clarify the Role of Section 5.1:**
   - What is the intended insight or hypothesis being tested in Section 5.1?
   - Can you show how this experiment validates or supports your theoretical findings?
   - **Suggestion:** Compute the optimal temperature for the real-data experiments using your closed-form expression (or an approximation), and compare it with performance trends in Section 5.1. Without this connection, the experiment appears disconnected from the main claims. It would also be interesting to see the comparison of a temperature based on the empirical variance  of `KZᵀ x QZ` in this experiment.


4. **Positioning within Existing Literature on Softmax Temperature:**
   - The importance of softmax temperature has been discussed in several works related to training dynamics, and ICL. L. 95 claims that its role in ICL has not been addressed, which appears to be incorrect.
   - **Suggestion:** Include a discussion of these related works.

[1] Z. Jiang, J. Gu and D. Z. Pan, "NormSoftmax: Normalizing the Input of Softmax to Accelerate and Stabilize Training," 2023 IEEE International Conference on Omni-layer Intelligent Systems (COINS), Berlin, Germany, 2023, pp. 1-6, doi: 10.1109/COINS57856.2023.10189242.

**Ethical Concerns:**

["NO or VERY MINOR ethics concerns only"]

**Final Justification:**

I'd like to thank the authors for this ***productive and helpful discussion***. I enjoyed this discussion phase and believe that discussion, clarification and additional experiments improved the submission quite significantly.

# Summary (for AC)

My main concerns have been addressed.

1) Most importantly, the role of section 5.2 has been strengthened and connected directly to the theoretical results. While initially the experiment was not directly related to the optimal temperature derivation this connection has now been added. This strengthens the theoretical results drastically, by showing a connection to a non-simplified setting. Unfortunately, this heuristic is limited by the need of choosing a scaling parameter. While this removes the practical value of the heuristic I am inclined to see over this limitation, as the main point of the paper is indeed the theoretical result and this experiment only empirical validation of this theoretical result. ***But, I ask the authors to mention this very prominently as a limitation in the revised draft***. Additionally, it would be important to highlight in the derivation that the heuristic is only proportional to the optimal value.

2) The connection between distribution shift and distribution of the QK dot product has not been made explicitly. Via the derivation of the above mentioned heuristic, this connection has now been made.

3) Authors promised to fix minor issues, mostly concerning presentation and relation to a broader set of related works.

# Final suggestions to authors

If authors can additionally run multiple experiments in different settings (for instance using different tasks or models) and find that using 1/2 as scale for the heuristic is always close to the optimum this would be ideal and would reduce concerns about practical value of the heuristic. This would allow to tone down the prominent discussion of limitations as requested above.

**Limitations:**

No, the authors should highlight that their theoretical result can (apparently) not be transferred to real data settings, where sources of noise are of different nature and not simple changes in mean, covariance, or Gaussian label noise.

**Quality:**

3

**Strengths And Weaknesses:**

### **Strengths**

- The paper is well written.
- It validates previous empirical findings by deriving a closed-form solution that explains the relationship between covariance, label noise, and the optimal temperature, a finding that is, to the best of my knowledge, novel.
- The paper confirms the theoretical results through experiments on linear regression ICL tasks.
- It highlights the importance of the attention temperature in transformer models for various tasks and training dynamics, as discussed in recent papers (which, however, are not referenced in this work).

---

### **Weaknesses**

- **Section 5.1 appears unrelated to the main contributions of the paper:**
  It is unclear what these experiments are intended to demonstrate or how they support the theoretical findings. Line 4 states that previous work has already established empirically that adjusting the temperature can improve ICL performance under distribution shifts (please provide an explicit citation here—presumably [35] or [8], but this should be clarified).
  The connection between these empirical results and the practical relevance of the theoretical findings is not clearly made. A possible improvement would be to estimate the optimal temperature in this setting using the closed-form solution, and highlight this value in the plot to show agreement between theory and empirical results.
  Without this link to the theoretical results, these experiments do not provide additional insight and would be better omitted.

- **The paper does not investigate how distribution shifts in the input data influence the input distribution of the attention softmax.**
  This relationship seems essential for understanding why modifying the temperature parameter can help mitigate poor generalization under distribution shifts.

- **Relatedly, it would be valuable to examine whether similar shifts in the softmax input distribution appear in the real-data experiments.**
  This would support the implicit claim that the behavior observed on synthetic tasks transfers to more complex, real-world settings.

---

### **Minor Weaknesses**

- **Figures 1–3:**
  The captions are purely descriptive and do not convey the key message of each plot. Strong captions should help the reader understand the figures even without reading the full paper.

- **Figure 3 uses 0.33 standard deviations as error bars.**
  This is a non-standard choice and raises questions about interpretability. Using full standard deviations as error bars would likely result in significant overlap between conditions, potentially undermining claims of significance. A statistical test (e.g., a t-test) would be more appropriate for assessing significance. I recommend to either use full standard deviation or standard error for the error bars.

- **Related work is not adequately discussed.**
  In particular, prior studies that explore the critical role of temperature in training and learning are not cited. The authors claim (line 95) that the effect of temperature on ICL has not been previously discussed. However, this claim would be more credible if the paper acknowledged and positioned itself with respect to relevant existing work.

---

> ### Author Rebuttal · Authors · 2025-07-31
>
> We sincerely thank you for your thoughtful review and detailed feedback. We appreciate your recognition of our paper's strengths, including the novelty of our closed-form solution for the optimal temperature, the empirical validation of our theoretical results, and the importance of temperature in in-context learning (ICL). We respectfully clarify that the core contribution of our work is the first theoretical characterization of the optimal attention temperature for ICL under distribution shift. Below, we address your concerns point by point.
>
> ### Regarding the role of Section 5.1-5.2
> > Section 5.1 appears unrelated to the main contributions of the paper: It is unclear what these experiments are intended to demonstrate or how they support the theoretical findings. $\cdots$
>
> > Clarify the Role of Section 5.1: What is the intended insight or hypothesis being tested in Section 5.1? Can you show how this experiment validates or supports your theoretical findings? $\cdots$
>
>
> We believe there may have been a misunderstanding regarding the section references. **Section 5.1 (Experiments on linear regression tasks)** is directly tied to our theoretical framework—it empirically validates our closed-form solution for the optimal attention temperature in the same controlled setting as our analysis. These experiments illustrate how distribution shifts in input, task, or label noise influence the optimal temperature, and confirm our theoretical predictions.
>
> However, we suspect your concern may have been aimed at **Section 5.2 (Experiments with LLMs on in-context QA tasks)**. The goal of Section 5.2 is to demonstrate the practical applicability of our theory to real-world scenarios. Specifically, it supports our central hypothesis: that appropriately adjusting attention temperature improves ICL performance under distribution shifts. While we do not compute a closed-form optimal temperature in this setting (due to complexity), these experiments provide empirical support for the generalizability of our theoretical insights. We agree that computing an approximation of the optimal temperature or overlaying theoretical predictions would make this link more explicit and plan to explore this in future work.
>
> ### Regarding the effect of distribution shift on the input of the softmax
>
> > The paper does not investigate how distribution shifts in the input data influence the input distribution of the attention softmax. This relationship seems essential for understanding why modifying the temperature parameter can help mitigate poor generalization under distribution shifts.
>
> > Relatedly, it would be valuable to examine whether similar shifts in the softmax input distribution appear in the real-data experiments. This would support the implicit claim that the behavior observed on synthetic tasks transfers to more complex, real-world settings.
>
>
> Thank you for raising this insightful point. While our primary focus is on how distribution shifts affect generalization error, we agree that analyzing the impact on the pre-softmax scores (i.e., $(KZ)^T(QZ)$) provides important complementary insight.
>
>
> Indeed, our theoretical derivation already involves the distribution of these dot products. In particular, **Theorem 4.6 leads to a closed-form expression for the optimal temperature (Eq. 15), where the numerator is a variance-like term and the denominator is a mean-like term** derived from the pre-softmax scores (i.e., $(KZ)^T(QZ)$). More concretely, we can rewrite optimal attention temperature as
> $$\tau_{optimal}
> =\frac{2 \times \text{(variance-like term)}}{\text{(mean-like term)}}$$
> where
>  - the **numerator** reflects the **second moment / variability** of $(KZ)^T(QZ)$, which increases under greater input covariance or label noise,
>
>  - the **denominator** reflects the **mean** of $(KZ)^T(QZ)$—the first-order statistics of softmax inputs.
>
> Thus, the optimal temperature balances variance against mean of softmax inputs. This provides a principled explanation for how and why the optimal temperature increases under distribution shifts that increase the variance of pre-softmax scores—a phenomenon observed in our experiments. We will include this interpretative discussion in the camera-ready version.
>
>
> > Clarify the Effect of Distribution Shifts on the Softmax Input Distribution: How exactly do changes in label noise or input covariance influence the mean and variance of the dot product $(K Z)^T (QZ)$? $\cdots$
>
>
> Empirically, increasing the input covariance or noise variance raises the variability of $(KZ)^T(QZ)$, which—as you pointed out—translates into an increased optimal temperature. This is consistent with both our theoretical predictions and experimental observations (Figure 2), thereby reinforcing the connection between distribution shift, variance, and optimal softmax temperature.
>
>
> > Relate Derived Optimal Temperature to Practical Baselines: Does your closed-form solution for the optimal temperature deviate significantly from the empirical variance of $(KZ)^T (QZ)$?
>
>
> Yes, it does. While practical heuristics such as using the empirical variance of $(KZ)^T(QZ)$ only consider the spread of softmax inputs, our closed-form solution accounts for both variance and mean, aiming to minimize generalization error rather than match a statistical moment. As such, our formula provides a more principled and task-aligned approach, especially under distribution shifts. We believe this refinement offers meaningful guidance beyond existing heuristics.
>
> ### Regarding the related work
> > Line 4 states that previous work has already established empirically that adjusting the temperature can improve ICL performance under distribution shifts (please provide an explicit citation here—presumably [35] or [8], but this should be clarified).
>
>
> Thank you for this observation. In line 4 (abstract), we intended to refer to prior empirical work that shows **performance benefits of tuning the attention temperature** (e.g., [16, 36, 21, 13, 5, 37]), but not in the specific context of ICL under distribution shifts. We agree that this statement could be clearer and will revise it for precision.
>
> > Related work is not adequately discussed.
>
> > In particular, prior studies that explore the critical role of temperature in training and learning are not cited. The authors claim (line 95) that the effect of temperature on ICL has not been previously discussed. However, this claim would be more credible if the paper acknowledged and positioned itself with respect to relevant existing work. $\cdots$
>
> > Positioning within Existing Literature on Softmax Temperature $\cdots$
>
>
> We appreciate this feedback. While prior work has discussed temperature in the context of training dynamics, calibration, or general transformer behavior, **to the best of our knowledge, our paper is the first to provide a theoretical characterization of the optimal attention temperature for ICL under distribution shift**. That said, we will revise the discussion to better acknowledge related work and clarify how our contribution builds on and extends these ideas. We will also include your suggested reference (e.g., [1]) and welcome any additional pointers.
>
> ### Regarding your recommendations about the figures
>
> > Figures 1–3: The captions are purely descriptive and do not convey the key message of each plot. Strong captions should help the reader understand the figures even without reading the full paper.
>
>
> Thank you—we agree. We will revise the captions for Figures 1–3 to explicitly communicate each plot's takeaway, improving standalone readability.
>
> > Figure 3 uses 0.33 standard deviations as error bars. This is a non-standard choice and raises questions about interpretability. Using full standard deviations as error bars would likely result in significant overlap between conditions, potentially undermining claims of significance. A statistical test (e.g., a t-test) would be more appropriate for assessing significance. I recommend to either use full standard deviation or standard error for the error bars.
>
>
> You're right. We used this for visual clarity, but recognize it's non-standard and may lead to misinterpretation. We will revise the figure to use full standard deviation and, where appropriate, supplement with statistical tests (e.g., t-tests) to establish significance.
>
> ---
>
> To address a common concern among reviewers regarding the practical value, in the camera-ready version of the paper, we will
>
> * **add a short Section "Selecting the attention temperature in practice"** that:
>
>   - (i) gives the **moment‑ratio heuristic** connecting moments of $(K Z)^T (Q Z)$ to the optimal attention temperature (Eq.15);
>
>   - (ii) includes the "variance vs. mean" explanation from the rebuttal into the paper, describing how increasing noise or covariance scale pushes the optimal temperature up in both the synthetic and LLaMA2 settings.
>
> ---
>
> Once again, thank you for your detailed and constructive feedback. Your comments have helped us clarify the contributions, improve the paper’s positioning, and enhance the presentation. We look forward to incorporating your suggestions in the camera-ready version and hope our clarifications positively impact your evaluation.

---

> > ### Comment · Reviewer_PaBX · 2025-08-04
> >
> > Thank you for your response and promised improvements.
> >
> >
> > **Regarding the role of Section 5.1-5.2**
> > Indeed, I intended to refer to 5.2. The connection of distribution shift and noisy labels needs to be made clearer. Why can we consider noisy labels as a distribution shift?
> >
> > The more important part, however, is the following:
> > In the rebuttal you state that the optimal temperature depends on the variance of $(KZ)^T(QZ)$ and the mean $(KZ)^T(QZ)$. So why not compute this for each attention head during the forward pass? This way you first, get the optimal temperature per head, which you can plot in Fig. 3 (a), but more importantly, you can directly use this value. If this agrees roughly with the best temperature found in Fig.3 (a) this is a strong validation of the theoretical results. Did I miss something or why did you not perform this experiment?
> >
> > If you can provide the above experiment and if possible compare to simply replacing the default temperature (expected variance) by the empirical variance for each sample I think this paper clearly shows that the theoretical results derived in a simplified setting are useful for realistic experiments and should be accepted. Without these experiments I am more doubtful and see this paper as a borderline case. In this case I would also suggest to drop section 5.2, as it does not relate to the theoretical findings for the optimal temperature. It's only an empirical finding that the default temperature is not optimal and the direction (larger/smaller temperature) as predicted by the insights.

---

> > > ### Author Response · Authors · 2025-08-04
> > >
> > > Thanks for your response and positive attitude towards our work.
> > >
> > > > The connection of distribution shift and noisy labels needs to be made clearer. Why can we consider noisy labels as a distribution shift?
> > >
> > > Thank you for requesting further clarification. In our LLM experiments, the models are originally trained on data that reflects correct scientific facts (e.g., the SCIQ dataset). At test time, we modify the labels provided in the prompt context by introducing incorrect or noisy information. This results in a mismatch between the distribution seen during training and the one encountered at inference — a classic instance of distribution shift.
> > >
> > > Conceptually, the noisy labels in the LLM experiments play a similar role to the injected label noise in our synthetic regression tasks. In both cases, the model is exposed to data distributions that deviate from its training distribution, particularly with respect to label reliability. This parallel motivates our comparison between the LLM experiments in Figure 3 and the synthetic experiments in Figure 2. In both real-world and synthetic settings, we observe a consistent trend: the optimal temperature increases as the level of label noise — and hence distribution shift — increases, reinforcing the robustness of our theoretical findings.
> > >
> > > > The more important part, however, is the following: In the rebuttal you state that the optimal temperature depends on the variance and the mean of $(K Z)^T (QZ)$. So why not compute this for each attention head during the forward pass? This way you first, get the optimal temperature per head, which you can plot in Fig. 3 (a), but more importantly, you can directly use this value. If this agrees roughly with the best temperature found in Fig.3 (a) this is a strong validation of the theoretical results. Did I miss something or why did you not perform this experiment?
> > >
> > > > If you can provide the above experiment and if possible compare to simply replacing the default temperature (expected variance) by the empirical variance for each sample I think this paper clearly shows that the theoretical results derived in a simplified setting are useful for realistic experiments and should be accepted.
> > >
> > > On the proposed experiment: We appreciate your insightful suggestion. We did not experiment with per-head or per-layer estimation of the optimal attention temperature, as it introduces additional computational overhead and complexity — particularly in robustly estimating the mean and variance at inference time across all layers and heads.
> > >
> > > However, we agree that this could strengthen our contributions from a practical standpoint. If time permits, we will conduct a targeted version of this experiment and include the results. Otherwise, we will incorporate a discussion of this idea in the final version and highlight it as an important direction for future work.

---

> > > > ### Comment · Reviewer_PaBX · 2025-08-05
> > > >
> > > > >  We did not experiment with per-head or per-layer estimation of the optimal attention temperature, as it introduces additional computational overhead and complexity.
> > > >
> > > > If I am not missing something here, I believe the computational cost is extremely low, i.e. it is a mean and variance computation per head x layers. In terms of implementation this should be at most 3 lines. Following this, I doubt the reason provided by the authors that computational complexity and time are truly limiting factors. If I am missing something please explain, otherwise, I believe there is no reason to not provide the results for this experiment. In the end, this is the missing link between theory and practice, that can show that the theoretical results are, despite simplifications, helpful.

---

> > > > > ### Author Response · Authors · 2025-08-07
> > > > >
> > > > > Thank you again for your thoughtful and constructive feedback. Following your suggestion, we conducted additional experiments to explore temperature selection using a moment-ratio heuristic. Below, we summarize our findings and how they align with our theoretical framework.
> > > > >
> > > > > ### 1. Per-head/layer temperature estimation
> > > > >
> > > > > As you proposed, we first attempted to derive per-head or per-layer temperature values using the moment-ratio heuristic (i.e., variance/mean). As anticipated, this approach proved difficult to apply in practice. Our theoretical framework (Eq. 15), defining an optimal temperature, does not directly say anything regarding per-head temperatures. Furthermore, moment estimation at the per-head level during inference turned out to be noisy and unstable, requiring additional heuristics and regularization that fall outside the scope of our current theoretical analysis.
> > > > >
> > > > > ### 2.  Global temperature derived from the moment-ratio heuristic
> > > > >
> > > > > Motivated by the above limitations, we turned to computing a single temperature value — as used in Fig. 3 — based on global statistics. Specifically, for each layer, we compute the mean of the diagonal entries and the variance of the off-diagonal entries of the pre-softmax attention scores. We then average these values over all layers and batches during inference to obtain a stable estimate, and set: $\tau = \frac{variance}{2 \times mean}$. This temperature is then applied uniformly across all heads and layers, consistent with the setup in our original Figure 3.
> > > > >
> > > > > Below, we report the performance of this new temperature choice alongside previously evaluated fixed temperatures.
> > > > >
> > > > > **Figure 3a: Results with temperature derived from the moment-ratio heuristic**
> > > > >
> > > > > Exact match scores (higher is better)
> > > > >
> > > > > | number of in-context examples -> |  1 |  2 |  3 |  4 |  5 | 6 | 7 | 8 |
> > > > > |:---:|:---:|:---:|:---:|:---:|:---:|:---:|:---:|:---:|
> > > > > | $\tau = 0.5$ | 12.75 | 18.92 | 16.67 | 16.75 | 16.08 | 15.50 | 14.67 | 12.58 |
> > > > > | $\tau = 0.8$ | 38.58 | 37.75 | 34.08 | 34.17 | 32.92 | 30.83 | 27.08 | 26.42 |
> > > > > | $\tau = 0.9$ | 44.75 | 41.33 | 41.08 | 36.58 | 37.50 | 32.08 | 28.67 | 29.00 |
> > > > > | $\tau = 1.0$ | 45.83 | 46.75 | 43.08 | 42.75 | 41.25 | 37.08 | 34.50 | 30.67 |
> > > > > | $\tau = 1.1$ | 43.75 | 50.25 | 47.33 | 46.25 | 44.25 | 41.75 | 38.17 | 33.67 |
> > > > > | $\tau = 1.2$ | 42.58 | 50.67 | 48.42 | 47.00 | 45.00 | 41.25 | 39.75 | 36.92 |
> > > > > | $\tau = 1.5$ | 21.25 | 29.17 | 30.17 | 27.83 | 28.42 | 28.17 | 25.33 | 24.83 |
> > > > > | $\tau = \frac{variance}{2 \times mean}$ **(new)** | **43.83** | **50.33** | **48.00** | **47.67** | **44.75** | **40.42** | **39.42** | **36.33** |
> > > > >
> > > > > **Figure 3b: Results with temperature derived from the moment-ratio heuristic**
> > > > >
> > > > > Exact match scores (higher is better)
> > > > >
> > > > > | noisy ratios -> |  0.01 |  0.20 |  0.40 |  0.60 |  0.80 | 0.99  |
> > > > > |:---:|:---:|:---:|:---:|:---:|:---:|:---:|
> > > > > | $\tau = 0.5$ | 25.00 | 22.50 | 18.08 | 15.50 | 11.25 | 8.17  |
> > > > > | $\tau = 0.8$ | 65.33 | 54.58 | 42.58 | 30.83 | 17.42 | 9.00  |
> > > > > | $\tau = 0.9$ | 67.33 | 57.75 | 46.67 | 32.08 | 20.92 | 10.00 |
> > > > > | $\tau = 1.0$ | 66.50 | 60.67 | 51.67 | 37.08 | 24.75 | 12.42 |
> > > > > | $\tau = 1.1$ | 66.42 | 59.50 | 51.83 | 41.75 | 27.33 | 15.75 |
> > > > > | $\tau = 1.2$ | 64.75 | 58.75 | 51.17 | 41.25 | 31.92 | 18.83 |
> > > > > | $\tau = 1.5$ | 39.75 | 36.33 | 32.33 | 28.17 | 20.50 | 15.92 |
> > > > > | $\tau = \frac{variance}{2 \times mean}$ **(new)** | **64.08** | **60.83** | **52.50** | **42.00** | **31.25** | **18.17**|
> > > > >
> > > > > We would like to note that since the moment-ratio heuristic just implies proportionality rather than equality, we empirically found that using a $1/2$ multiplier (in front of the moment-ratio) to be useful in our experiments.
> > > > >
> > > > > ### Clarification on theoretical connection:
> > > > >
> > > > > We would also like to reiterate that while the moment-ratio heuristic is inspired by our theoretical formulation (Eq. 15), it does not capture the full complexity of the optimal temperature. The theoretical expression involves the exact mean vectors, covariance matrices, and vector/matrix operations on them. Thus, the optimal temperature (Eq. 15) is not directly determined by the empirical mean and variance of pre-softmax scores. Nevertheless, the success of this heuristic further supports the practical relevance of our theoretical insights.
> > > > >
> > > > >
> > > > > ### Conclusion
> > > > > These results demonstrate that, while our theory does not explicitly prescribe per-head temperature scaling, the global moment-ratio heuristic — inspired by the theoretical formulation in Eq. 15 — leads to strong empirical performance across varying numbers of in-context examples and noise levels. We believe this addresses the core of your suggestion and shows that the theoretical insights can, in practice, guide robust temperature selection in large language models.
> > > > >
> > > > > We again thank you for this insightful suggestion, which helped us improve both the clarity and practical relevance of our work. We hope that these clarifications and additional results positively contribute to your final evaluation.

---

> > > > > > ### Comment · Reviewer_PaBX · 2025-08-07
> > > > > > **Mandatory**
> > > > > >
> > > > > > Thanks for the additional experiments and clarifications. I increased my score accordingly. Please see "Final Justification" for more details and further suggestions.

---

### Official Review · Reviewer_7yS3 · 2025-07-01

**Clarity:** 3
**Significance:** 3
**Originality:** 3
**Rating:** 4
**Confidence:** 4

**Summary:**

This work theoretically analyzes the performance of ICL using a linearized attention (with row-wise normalization), under a linear regression setting with shifted mean values of task weights. Based on the theory, this work analyzes how different kinds of distribution shifts would affect the ICL performance and empirically validates the analysis. The authors further theoretically derive an optimal temperature. The experiments on an Llama-2 demonstrate the effect of temperature on the ICL performance on a noisy label task.

**Questions:**

I'm willing to increase my score if the authors could address some of the following concerns:

Could you provide a guideline on how to choose an appropriate temperature for real LLMs? Otherwise, this paper would be less valuable in real applications.
Could you elaborate on how the theoretically optimal temperature can compensate the generalization error under your simplified theoretical setting? For example, you may manually construct some OOD tasks following the cases in Sec. 4.2 and compute the reduced quality of the generalization error by using the optimal temperature. Or, at least, you could intuitively and systematically show how adjusting the temperature will improve generalization under different kinds of distribution shifts.
Could you include more real-world LLM experiments? From my point of view, designing or finding what are "OOD" tasks for real-world LLM can also be contributive.

**Ethical Concerns:**

["NO or VERY MINOR ethics concerns only"]

**Final Justification:**

Thanks to the authors for the detailed response and experimental results, which are great additions to the paper.

**Limitations:**

No. My suggestions are mainly included in the question part.

**Quality:**

3

**Strengths And Weaknesses:**

Strengths:

Compared to previous theoretical works, this work considers a slightly more general setting (although still a linear regression task) where the task weights w
 can have non-zero mean μw
.
This work considers an attention architecture with row-wise normalization, which is slightly more general than the linear attention adopted by some of the previous works on ICL theory, like [2].
The theoretical result provides insights into how the distribution shifts in different components affect ICL performance. The simulated experiments (Fig. 1 and 2) align well with the discussions based on the theory (Sec. 4.2).
Weaknesses:

My main concern lies in the practical application value of this article in real tasks. The authors don't offer a guideline (even just a vague one) in how to select a better temperature for improving the generalization of ICL, nor do they theoretically reveal how the temperature could handle each types of distribution shifts mentioned in Sec. 4.2 (please see Question 2). Moreover, it's not surprising that tuning the temperature could help to improve the performance for LLMs. There seems to lack more concrete theoretical predictions of the effect of the temperature besides Fig. 3.
This work considers a simplified single-layer linearized transformer, with the unrealistic (although widely adopted) embedding structure (Eq. 4). The limitations of such hand-constructed transformers have already been pointed out by [1]. Also, although being more general than the linear attention, I still doubt whether the linearized attention (normalized) considered in this work can truly reflect the dynamics of ICL by real-world LLMs. Especially, there have been lots of works conducting theoretical analysis on more complex nonlinear transformers ([3] [4] [5]).
The generality of distributional shifts considered in the theory is limited. The main distribution shift arises from the changes in the parameters of the same linear regression task (although the changes in the mean of the task weights make it somewhat more general). However, real-world distribution shifts, especially for LLM tasks, can be various. Whether the results about generalization can be applied to larger distribution shifts remains questionable. Anyway, I understand this is for the ease of analysis, and it's challenging to provide for arbitrary distribution shifts, so this would be a minor concern.
The experiments for real-world LLMs is constrained to a noisy label task.
[1] Do pretrained Transformers Learn In-Context by Gradient Descent? ICML 2024

[2] How do Transformers perform In-Context Autoregressive Learning? ICML 2024

[3] In-Context Learning with Transformers: Softmax Attention Adapts to Function Lipschitzness. NeurIPS 2024

[4] How Do Nonlinear Transformers Learn and Generalize in In-Context Learning? ICML 2024

[5] Towards Better Understanding of In-Context Learning Ability from In-Context Uncertainty Quantification. arXiv:2405.15115

---

> ### Author Rebuttal · Authors · 2025-07-31
>
> Thank you for your detailed review and constructive feedback, particularly your request for additional practical guidance regarding the choice of the attention temperature. We also appreciate your recognition of the strengths of our work—namely, the general theoretical framework, the insights into the relationship between distribution shifts and in-context learning (ICL) performance, and the accompanying experimental validation.
>
> We would like to emphasize that the primary contribution of our work lies in providing the first complete theoretical characterization of attention temperature in ICL under distribution shifts. This represents a novel and original direction that, to our knowledge, has not been explored in prior literature. We address each of your concerns and questions below.
>
>
> ### Regarding the practical value
> > My main concern lies in the practical application value of this article in real tasks. $\cdots$
>
> Thank you for your feedback. First and foremost, as a theoretical contribution, our paper does not primarily aim to offer practical guidelines. Rather, it seeks to provide an analytical understanding of the role of attention temperature in the in-context learning (ICL) performance of Transformers under distribution shifts. That said, in light of your comments, we have developed a high-level intuition that may inform the practical selection of attention temperature. This is discussed in more detail below, under the question regarding practical guidelines.
>
>
> > Could you elaborate on how the theoretically optimal temperature can compensate the generalization error under your simplified theoretical setting? $\cdots$
>
> Thank you for requesting clarification regarding the relationship between the optimal attention temperature and distribution shifts. Indeed, Figures 1 and 2 (discussed in Section 5.1) experimentally illustrate how selecting the optimal temperature improves generalization performance under the distribution shift scenarios presented in Section 4.2. Specifically, Case I (shift in the input distribution) is depicted in Figure 1b; Case II (shift in the task distribution) is shown in Figure 1c; and Case III (shift in the noise distribution) is examined in Figure 2. Each of these figures quantitatively demonstrates how an appropriately chosen temperature enhances the in-context learning (ICL) performance of Transformers under various types of distribution shifts. We will clarify this connection more explicitly when introducing the cases in Section 4.2.
>
>
> To further elaborate on the mechanism by which the optimal temperature (Eq.15) improves the generalization error in in-context learning (ICL) under distribution shifts, we establish a connection between the closed-form expression for the optimal temperature (Eq.15) and the nature of the distribution shifts. This analysis also leads to a practical guideline, presented below.
>
>
> > I'm willing to increase my score if the authors could address some of the following concerns: $\cdots$
>
>
> Thank you for your feedback and for your constructive intentions. To enhance the practical value of our work, we analyze the relationship between the input to the softmax function in the attention mechanism and the optimal temperature identified in Theorem 4.7, since the attention temperature effectively acts as a scaling factor for the softmax input in practice.
>
> Specifically, we establish a connection between the statistical moments of the dot product $(KZ)^T (QZ)$ (the input to the softmax) and the optimal temperature (Eq.15), which can guide the practical selection of the attention temperature.
>
> First, the numerator of the optimal temperature (Eq.15) corresponds to a variance-like term, aggregating the **second moment / variance** of the pre-softmax scores (i.e., the elements of $(KZ)^T (QZ)$) induced by the test-time distribution parameters $(\Sigma_x, \sigma^2)$ and the learned parameters $(M_{11}, v_{21}, v_{22})$. Intuitively, greater variability in the elements of $(KZ)^T (QZ)$ increases the numerator, thereby raising the optimal temperature.
>
> Conversely, the denominator of the optimal temperature (Eq.15) is a mean-like term, collecting the **first-order** components from the same building blocks and capturing the **mean** of $(KZ)^T (QZ)$ that is useful for prediction. A Stronger mean-like term lowers the optimal temperature.
>
> Thus, the optimal temperature balances **variance** against **mean**, which aligns with the intuition of “right-sizing” inputs before a softmax operation. In our experiments, increasing either the input covariance or noise variance raises the variability of $(KZ)^T (QZ)$, which corresponds to an increased optimal temperature.
>
> We will include a detailed discussion of this relationship, emphasizing its implications for the practical selection of attention temperature, in the camera-ready version of the paper. We again thank you for your valuable feedback.
>
>
> > Moreover, it's not surprising that tuning the temperature could help to improve the performance for LLMs. There seems to lack more concrete theoretical predictions of the effect of the temperature besides Fig. 3.
>
> Thanks for the feedback. The temperature does not always improve the performance for LLMs. What we demonstrate in this paper is that the optimal temperature depends on the nature of the distribution shifts. For instance, increasing the noise level or input scaling leads to greater variability in $(KZ)^T (QZ)$, which in turn raises the optimal temperature. Figure 3b (the LLM experiment with noisy in-context examples) specifically confirms this relationship in a real-world scenario.
>
> > Could you include more real-world LLM experiments? $\cdots$
>
>
> Thank you for your valuable suggestions for future work. We agree that additional experiments with real-world LLMs and a deeper characterization of out-of-distribution (OOD) tasks would greatly benefit the field. However, these directions are beyond the scope of the current paper, which focuses primarily on theoretical understanding rather than immediate practical applications. Consequently, we have left the translation of the theoretical insights presented here into directly applicable practical methods for future work.
>
>
> ### Regarding the setting
>
> > This work considers a simplified single-layer linearized transformer, with the unrealistic (although widely adopted) embedding structure (Eq. 4). $\cdots$
>
>
> Thank you for your comment regarding our experimental setting. We agree that extending our work to more realistic scenarios is an important and promising direction for future research. However, we would like to emphasize that our current setting is a deliberate choice that balances analytical tractability with the representativeness of the effects of attention temperature on in-context learning (ICL) under distribution shifts.
>
> Specifically, while the theoretical studies you mentioned investigate more complex nonlinear Transformers, they do not provide asymptotic characterizations of the generalization error for ICL—such as our Theorem 4.6—which enable us to derive a closed-form expression for the optimal attention temperature (Theorem 4.7). For these reasons, we believe our current setting is both useful and sufficiently expressive for studying attention temperature in ICL under distribution shifts.
>
>
> ### Regarding the distribution shifts
> >  The generality of distributional shifts considered in the theory is limited. The main distribution shift arises from the changes in the parameters of the same linear regression task (although the changes in the mean of the task weights make it somewhat more general). $\cdots$
>
> Thank you for the opportunity to discuss distribution shifts. As with most theoretical work, we begin with clear assumptions about the data model and perform our analysis within this framework. In the paper, we consider and discuss all possible distribution shifts within our data model, specifically shifts in the input, task, or noise distributions.
>
> We agree that extending these results to more general data settings would be valuable, but as you noted, it is quite challenging to carry out a rigorous asymptotic analysis in such broader contexts. Nonetheless, we believe our LLM experiments demonstrate that the insights regarding the optimal attention temperature extend beyond the theoretical setting considered here.
>
> Therefore, guided by the high-level intuition described above, we anticipate that future work can translate our theoretical findings into practical value for more general scenarios.
>
>
> ---
>
> To address a common concern among reviewers regarding the practical value, in the camera-ready version of the paper, we will
>
> * **add a short section "Selecting the attention temperature in practice"** that:
>
> * - (i) gives the **moment‑ratio heuristic** connecting moments of $(K Z)^T (Q Z)$ to the optimal attention temperature (Eq.15);
>
> * - (ii) includes the "variance vs. mean" explanation from the rebuttal into the paper, describing how increasing noise or covariance scale pushes the optimal temperature up in both the synthetic and LLaMA2 settings.
>
> ---
>
>
> Finally, we would like to emphasize the importance and originality of our asymptotic analysis of the attention temperature for ICL under distribution shifts, a topic that has been largely overlooked in prior theoretical literature. We hope that our clarifications addressing your concerns, as well as our explanations regarding the practical implications of our work, will help guide your evaluation toward a more favorable rating.

---

> > ### Comment · Reviewer_7yS3 · 2025-08-05
> >
> > Thanks to the authors for the detailed response and experimental results, which are great additions to the paper. I have no further questions.

---

> > > ### Author Response · Authors · 2025-08-07
> > >
> > > Thank you for the positive response. We’re glad to hear that you found our additions to be valuable and that you have no further questions. We hope that our clarifications and additions have positively contributed to your overall assessment of the paper.

---

### Official Review · Reviewer_1PUu · 2025-07-02

**Clarity:** 3
**Significance:** 3
**Originality:** 3
**Rating:** 4
**Confidence:** 3

**Summary:**

This paper investigates the role of attention temperature in improving in-context learning (ICL) robustness of pretrained Transformers under distribution shifts. By analyzing a simplified framework with linearized softmax attention, the authors derive a closed-form expression for the optimal temperature that minimizes generalization error, showing its dependence on the nature of distribution shifts (e.g., input covariance or label noise). Theoretical insights are validated through experiments on synthetic linear regression tasks and real-world question-answering benchmarks with LLaMA2-7B, demonstrating that tuning temperature effectively mitigates performance degradation under distribution shifts.

**Questions:**

None

**Ethical Concerns:**

["NO or VERY MINOR ethics concerns only"]

**Final Justification:**

This is an interesting paper, but it would be strengthened by additional experiments. Therefore, I ​​maintain​​ my original rating: ​​4​​.

**Limitations:**

Please refer to the weaknesses section.

**Quality:**

2

**Strengths And Weaknesses:**

Strengths:

1. The paper provides the first theoretical analysis of attention temperature in ICL under distribution shifts, filling a critical gap in understanding how temperature modulates attention weights to improve generalization.
2. Using linearized softmax attention balances mathematical tractability with realism, retaining temperature-dependent behaviors of standard attention while enabling closed-form derivations for optimal temperature.
3. The findings are consistently supported by both controlled synthetic experiments (linear regression) and large-scale LLM tests (LLaMA2-7B on SCIQ), enhancing credibility across model scales and tasks.

Weaknesses:

1. The focus on linearized softmax attention, while analytically useful, may limit direct generalization to standard multi-head attention in modern Transformers, which involves more complex interactions.
2. The theoretical analysis is centered on linear regression, and while extended to QA, broader validation on diverse tasks (e.g., classification, sequence generation) would strengthen generalizability.
3. The optimal temperature derivation relies on knowing distribution shift properties (e.g., covariance changes), which may be unavailable in real-world scenarios, raising questions about its applicability without prior knowledge of shifts.

---

> ### Author Rebuttal · Authors · 2025-07-31
>
> Thank you for your comprehensive review and positive feedback. We sincerely appreciate your recognition of the strengths of our paper---namely, the first theoretical analysis of attention temperature in ICL under distribution shifts, the use of **linearized softmax attention** to balance mathematical tractability with realism, and the experimental confirmation of our theoretical findings. We view the points you raised under weaknesses as valuable suggestions for future work, especially given the theoretical focus of our paper, and we agree with them in that context.
>
> That said, considering the strengths you acknowledged, we believe our work merits a rating higher than borderline. We hope that after reviewing our clarifications below, you will consider raising your evaluation. We respond to each of your concerns (listed under weaknesses) as follows:
>
>
> > The focus on linearized softmax attention, while analytically useful, may limit direct generalization to standard multi-head attention in modern Transformers, which involves more complex interactions.
>
> Thank you for this feedback. We agree that extending the analysis to more general settings (e.g., multi-head attention in modern Transformers) is important. In this direction, we present large language model (LLM) experiments (Figure 3) involving the LLaMA2-7B model on the SCIQ dataset, which indicate the usefulness of the optimal attention temperature in real-world problems. These experimental results also connect meaningfully with our theoretical findings.
>
> Specifically, our analysis reveals that the optimal temperature increases with the noise level---as shown in Figure 2b for the theoretical setting and Figure 3b for the LLM experiments. This finding is consistent with our closed-form expression for the optimal temperature (Equation 15). Here, the statistical moments of the dot product $(KZ)^T(QZ)$---the input to the softmax---are directly related to the optimal temperature, offering practical guidance for its selection.
>
> In particular, the **numerator** of the optimal temperature in Equation 15 is a variance-like term. It aggregates the **second moment / variance** of the pre-softmax scores (i.e., the elements of $(KZ)^T(QZ)$), which are influenced by the test-time distribution ($\Sigma_x$, $\sigma^2$) and learned parameters ($M_{11}$, $v_{21}$, $v_{22}$). Intuitively, greater variability in the elements of $(KZ)^T(QZ)$ increases the numerator, thereby increasing the optimal temperature.
>
> Meanwhile, the **denominator** of the optimal temperature is a mean-like term, reflecting the **first-order** components in the same factors. It captures the **mean** of $(KZ)^T(QZ)$ that contributes positively to prediction accuracy. A stronger mean-like term decreases the optimal temperature.
>
> Thus, the optimal temperature embodies a balance between **variance** and **mean**, aligning with the intuition behind appropriately scaling logits before applying a softmax.
>
> We will include a discussion of this relationship---with an emphasis on the practical selection of attention temperature---in the camera-ready version of the paper. Once again, thank you for the valuable feedback.
>
>
> > The theoretical analysis is centered on linear regression, and while extended to QA, broader validation on diverse tasks (e.g., classification, sequence generation) would strengthen generalizability.
>
> Thank you for this comment. While we agree that broader validation across diverse tasks (e.g., classification, sequence generation, etc.) would further strengthen our findings, we believe that the current results are sufficient given the theoretical focus of our work. Accordingly, we view the extension of our experiments to a wider range of settings as a promising direction for future work with a more empirical emphasis.
>
>
> > The optimal temperature derivation relies on knowing distribution shift properties (e.g., covariance changes), which may be unavailable in real-world scenarios, raising questions about its applicability without prior knowledge of shifts.
>
> Thank you for this point. We agree that the optimal temperature derived with knowledge of distribution shift properties cannot be directly applied in real-world scenarios.
>
> However, we would like to clarify that, as a theoretical contribution, our paper aims to explain the relationship between distribution shifts and the optimal attention temperature. This leads to a deeper theoretical understanding of the role of attention temperature in in-context learning (ICL) under distribution shifts.
>
> Therefore, while the optimal temperature itself may not be directly usable in practice, our findings---including the closed-form expression for the optimal temperature---offer practical insights into how attention temperature should be selected in real-world settings (as discussed above).
>
>
> ---
>
> To address a common concern among reviewers regarding the practical value of our work, we plan to incorporate the following updates in the camera-ready version of the paper:
>
>
> * **Add a short section "Selecting the attention temperature in practice"** that:
>     - (i) Presents the **moment-ratio heuristic**, which connects the statistical moments of $(KZ)^T(QZ)$ to the optimal attention temperature (Equation~15).
>
>     - (ii) Integrates the **variance vs. mean** explanation from the rebuttal, describing how increasing the noise level or covariance scale raises the optimal temperature in both synthetic and LLaMA2-based experimental settings.
>
> ---
>
> Thank you again for your thoughtful review and positive evaluation. Your support is especially important to us, as many of the other reviews have focused more heavily on practical considerations, sometimes at the expense of recognizing the theoretical contributions of our work.
>
> In this context, we greatly appreciate your acknowledgment of the strengths of our paper. We hope that our clarifications regarding the listed weaknesses address your concerns and will encourage you to consider raising your rating.

---

> > ### Comment · Reviewer_1PUu · 2025-08-06
> >
> > My concerns have been addressed. I will keep my score.

---

> > > ### Author Response · Authors · 2025-08-07
> > >
> > > Thank you for your response. We’re glad to hear that your concerns have been addressed. Based on this, if you reconsider your rating, we would greatly appreciate it. Please don’t hesitate to let us know if you have any further questions or feedback.

---

### Official Review · Reviewer_T7Mg · 2025-07-03

**Clarity:** 3
**Significance:** 1
**Originality:** 3
**Rating:** 4
**Confidence:** 4

**Summary:**

This paper provides novel theoretical analysis on in-context learning particularly in context of attention temperature under distribution shifts. By simplifying the attention-mechanisms in transformers into linearized softmax attention, this paper shows optimal softmax temperature can reduce the train-test error comparable to optimal bayes. This extends to more complex transformers in real-world tasks that the optimal temperature varies across distribution shifts.

**Questions:**

Please see above.

**Ethical Concerns:**

["NO or VERY MINOR ethics concerns only"]

**Final Justification:**

Since the authors' rebuttal have addressed some of my concerns, I'd like to raise my rating to 4.

**Limitations:**

Please see above.

**Paper Formatting Concerns:**

No.

**Quality:**

3

**Strengths And Weaknesses:**

**Strengths**
* This paper provides a novel theoretical perspective of the attention softmax temperature as a variable that fixes models' generalizations error given distribution shifts, which was under-explored in the field.
* Its theory extends to more complicated and real-world cases, e.g., LLaMA-7B, supporting the authors' claim regarding the existence of optimal temperature for distribution shifts.

**Weaknesses**
* I'm not fully convinced how important this paper's finding is to community's understanding on ICL beyond the theoretical understanding of optimal temperature for robust ICL. What aspects the authors expect the future directions of research this paper might inspire?
* Practical implication is not clear: Even if this paper confirms the empirical existence of optimal temperature in more complex models, it remains the same regarding how to tune the optimal temperature as in conventional way, i.e., tuning it using validation set on test data. Therefore, I believe it lacks the practical implication while it claims that it offers the practical guidance for tuning Pretrained Transformer.

---

> ### Author Rebuttal · Authors · 2025-07-31
>
> Thank you for your detailed review of our work. We appreciate your recognition of the novel theoretical perspective introduced in our paper regarding attention temperature, as well as the real-world experiments that support our findings. Below, we address the concerns you raised.
>
> ### Regarding the future directions
>
> > I'm not fully convinced how important this paper's finding is to community's understanding on ICL beyond the theoretical understanding of optimal temperature for robust ICL. What aspects the authors expect the future directions of research this paper might inspire?
>
>
> Thank you for the opportunity to elaborate on this. We believe that attention temperature plays a critical role in scenarios involving distribution shift. From a practical perspective, our demonstration of the existence and benefits of an optimal attention temperature opens the door to more principled methods for tuning or adapting temperature in real-world applications. For instance, promising future directions include:
>
> - Developing adaptive mechanisms for selecting attention temperature based on test-time statistics.
>
> - Investigating the layer-wise or head-specific adjustment of temperature for improved robustness.
>
> From a theoretical standpoint, our results suggest that attention temperature is not merely a tuning hyperparameter, but a core component that governs the success of ICL under distribution shift. We hope this encourages future work to explicitly account for temperature when analyzing ICL with Transformers, particularly in the presence of mismatched training and test distributions—an aspect often overlooked in current theoretical treatments.
>
> ### Regarding the practical implications
>
> > Practical implication is not clear: Even if this paper confirms the empirical existence of optimal temperature in more complex models, it remains the same regarding how to tune the optimal temperature as in conventional way, i.e., tuning it using validation set on test data. Therefore, I believe it lacks the practical implication while it claims that it offers the practical guidance for tuning Pretrained Transformer.
>
>
> Thank you for this important feedback. In response to this and similar comments, we revisited our theoretical and empirical findings and derived a **high-level practical insight**:
> There exists a **direct connection between the statistical moments of the pre-softmax scores**, i.e., the dot products $(KZ)^T(QZ)$, and the optimal attention temperature. This insight provides a **theoretically grounded heuristic** for guiding temperature selection in practice. Specifically:
>
> * The **numerator** of the optimal temperature (Eq.15) is a **variance-like term** that aggregates the **second moment** of $(KZ)^T(QZ)$, influenced by the test-time distribution $(\Sigma_x, \sigma^2)$ and the learned parameters $(M_{11}, v_{21}, v_{22})$. Intuitively, **higher variability** in these dot products raises the numerator and thus increases the optimal temperature.
> * The **denominator** is a **mean-like** term that captures the **signal strength** in the same components, promoting sharper predictions. A stronger mean-like term reduces the optimal temperature.
>
>
> Hence, the optimal temperature effectively **balances prediction mean and variance**—a principle that directly reflects how one might intuitively “right-size” logits before a softmax.
>
> We will incorporate this insight in the camera-ready version to enhance the paper's practical utility.
>
>
> ---
>
> To further address the recurring concern about real-world applicability, we will
>
> * **include a short section "Selecting the attention temperature in practice"** that:
>     - (i) Present a **moment-ratio heuristic** linking the variance and mean of $(KZ)^T(QZ)$ to the optimal attention temperature (Eq.15).
>     - (ii) Include the "variance vs. mean" explanation discussed above, and show how **noise and input covariance shifts** influence the optimal temperature in both synthetic and LLaMA2 settings.
>
> ---
>
> Once again, we sincerely thank you for your thoughtful comments and the time you invested in reviewing our work. We hope these clarifications strengthen your assessment of our contribution.

---

> > ### Comment · Reviewer_T7Mg · 2025-08-05
> >
> > I appreciate the authors' responses to my concerns.
> >
> > I believe the responses have clarified the practical implications on importance of temperature for generalizations of ICL under distribution shifts. Still, as other reviewers noted as well, I am still not fully convinced how important the optimal temperature is beyond the simple setup and how generalizable it would be to the real-world LLMs' generalizations.
> >
> > Therefore, I would like to raise my rating to 4.

---

> > > ### Author Response · Authors · 2025-08-07
> > >
> > > Thank you for your thoughtful feedback and for raising your rating. Regarding your concern about the generalizability of our findings to other settings, we believe that our experiments with large language models (as shown in Fig. 3) provide supporting evidence for the broader applicability of our approach. In response to a request from the reviewer PaBX, we have further extended this setting, and the updated results continue to support our conclusions.

---

### Decision · Program_Chairs · 2025-09-17

**Decision:**

Reject

**Comment:**

The paper studies the effect of the temperature in self-attention models, especially when testing in out-of-distribution performance. The paper studies self-attention layers trained on constructed linear regression tasks, when varying mean and covariance during test-time.

Strengths: All reviewers including myself find the study of temperature of attention in ICL interesting, and novel to study in constraint settings.

Weaknesses: All reviewers, although interesting, find the paper not well-motivated and are unsure about the
1) practical implications in terms of setting temperatures 2) shift to real world tasks and most importantly to me 3) theoretical tranfer to softmax self-attention.

After reading the work, I find the paper misleading as the temperature in linear self-attention is not well defined (and shouldn't be called temperature). As the paper is building on top of many works that study the linearized softmax setting, and provide evidence of transformer to the softmax case, this is somewhat understandable. Nevertheless, I find the transfer of temperature changes in terms of generalization (in theoretical settings as well as practical) not well discussed, and only shown in the last real world setting, potentially introducing many confounders.

From a high level: Temperature in softmax controls the peakyness of the distribution, and doesnt allow for "generalization" in the limit of temperature to 0, as we recall a certain training example. This is not at all the behaviour for linear self-attention, here all examples are scaled with the same scalar. I find this not at all addressed in the paper.
This paper is borderline, but I find the reviews quite shallow and none of the reviewers are addressing this crucial point - therefore I want to overrule the overall decision of borderline acceptance and reject the paper.